

# The World aerosol Optical depth Research and Calibration Center (WORCC), Quality assurance and quality control of GAW-PFR AOD measurements

Stelios Kazadzis[1], Natalia Kouremeti[1], Stephan Nyeki[1], Julian Gröbner[1], Christoph Wehrli[1]

[1]Physikalisch-Meteorologisches Observatorium Davos, World Radiation Center (PMOD/WRC) Dorfstrasse 33, CH-7260 Davos Dorf, Switzerland

*Correspondence to*: Stelios Kazadzis (stelios.kazadzis@pmodwrc.ch)

**Abstract.**

The World Optical Depth Research Calibration Center (WORCC) is a section within the World Radiation Center at Physikalisches-Meteorologisches Observatorium (PMOD/WRC), Davos, Switzerland, established after the recommendations of WMO for calibration of AOD related sun-photometers. WORCC is mandated to develop new methods for instrument calibration, to initiate homogenization activities among different AOD networks and to run a network (GAW-PFR) of sun-photometers. In this work we describe: the calibration hierarchy and methods used under WORCC and the basic procedures,

test and processing techniques in order to ensure the quality assurance and quality control of the AOD retrieved data.

## 1 Introduction

Aerosols in the atmosphere, through the direct and indirect effect, mainly result in a cooling contribution to the global radiation balance (IPCC, 2013). The parameter that describes their integrated optical attenuation is the aerosol optical depth (AOD) that can be derived by measurements of the sunlight transmittance (WMO, 2016). AOD has been used in case/local

studies in order to characterize aerosols, assess atmospheric pollution and the aerosol related radiative forcing. For these reasons, AOD has been measured with the use of sun-photometers for more than 50 years (Holben et al., 1998). Most of these measurements are site-specific, with little relevance to long term trend analysis on a global scale, however, several multi-year spatial studies (Holben, 2001; Che et al., 2015, Mitchell et al., 2017) have been conducted.

The World Meteorological Organization (WMO) instigated the Global Atmosphere Watch (GAW) program in 1989 as a

successor to the Background Air Pollution Monitoring Network (BAPMoN). In 1993 (WMO, 1993) it was recommended that AOD measurements, conducted previously under BAPMoN, should be discontinued until new instruments, methods and protocols could be established to collect AOD data of known and assured quality. Based on a recommendation by GAW experts, the World Optical Depth Research Calibration Center (WORCC) was established in 1996 at the PMOD/WRC in Switzerland. WORCC has since been advised by the GAW Scientific Advisory Group for Aerosols. Fifteen existing GAW

stations were chosen for the deployment and operation of 12 "N" type Precision Filter Radiometers (PFR; manufactured by PMOD/WRC) (Wehrli, 2000), provided by the Swiss Government. WORCC was assigned the following tasks:

- Development of a radiometric reference for spectral radiometry to determine AOD.
- Development of procedures to ensure worldwide homogeneity of AOD observations.
- Development of new instrumentation and methods for AOD.

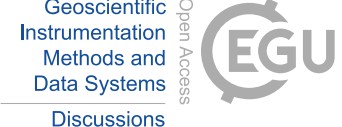

- Implementation of a pilot network for AOD at GAW global observatories including quality control and quality assurance of data, called GAW-PFR.

- Training operators to use and maintain AOD instruments.

There are different global networks measuring AOD, mainly distinguished by the different instruments used in each of them.

The aerosol robotic network (AERONET), (Holben et al., 1998; 2001) (http://aeronet.gsfc.nasa.gov/) is the major global network with central calibration facilities in the USA, France and Spain. The sky radiometer network (SKYNET) aerosol network (Takamura and Nakajima, 2004) is an observational network dedicated to aerosol-cloud-radiation interaction research studies. The Australian AOD network includes 22 stations (Mitchell et al., 2017). The Chinese Aerosol Remote Sensing network (CARSNET), reporting AOD measurements for 50 sites, representing remote, rural, and urban areas (Che

et al., 2015). In addition, national, regional and global networks such as the French component of AERONET, PHOTONS (Goloub et al., 2007), the Iberian Network for aerosol measurements (RIMA) (Prats et al., 2011), Aerosol Canada (AEROCAN), (Bokoye et al., 2001), have contributed to AOD climatology studies.

The Swiss-built PFR (Wehrli in WMO, 2005) has been operating continuously at 15 GAW stations and another 23 associated ones, worldwide. The PFR is expressly designed to make automated long-term observations at four wavelengths

(368, 412, 500 and 862 nm). Several studies using data from the GAW-PFR network have been published mainly focusing on long term changes of AOD (e.g. Ruckstuhl et al., 2008, Nyeki et al., 2012; 2015). In addition to these studies, GAW-PFR aims to provide inter-comparison information between networks by overlapping at selected sites.

PFR instruments of the GAW-PFR Network currently overlap with AERONET, SKYNET, CARSNET, the Australian Network and other sun-photometers at several sites. As there is a need for a common strategy to merge the various network

observations into a global data set, the WMO-GAW scientific advisory group for aerosols recommended that GAW will have to collaborate with existing major networks to develop this strategy, implementing and developing, together with satellite agencies, a system for integrating global AOD observations. Towards the same goal, WORCC organizes a filter radiometer comparison every five years with the participation of a number of reference AOD measuring instruments from different networks. The last comparison was held in 2015 with the participation of 30 instruments (WMO, 2016). In this

work, we present the research activities of WORCC and more specifically the calibration hierarchy, the quality assurance and quality control of the GAW-PFR network AOD measurements.

The characterization, calibration of the PFR instruments together with the quality assurance and quality control of the measurements is WORCC's major task. In this work, we summarize the calibration procedures and hierarchy since the GAW-PFR network was established, 17 years ago and the quality assurance and control of the data that the instruments

provide.

## 2 Calibration principles and hierarchy

AOD is a dimensionless quantity that cannot be measured directly. It can be retrieved from atmospheric transmission measurements and cannot be directly linked to any SI (International System of Units) reference since the atmospheric transmission is also a relative factor related to: the direct solar irradiance (I) at a particular wavelength ($\lambda$), and I($\lambda$) at the

surface and at the top of the atmosphere $I_o(\lambda,r)$, where r is the Sun-Earth distance. As a consequence, transmission can be measured in any units and in the case of sun-photometers, the instrument voltage signal V($\lambda$) and the signal at the top of the atmosphere (extra-terrestrial value $V_o(\lambda,r)$), can be used for the AOD determination using the Beer-Lambert law:

$$\tau_{aer} = \frac{ln(Vo) - ln(V)}{m} - \tau_{rt} \tag{1}$$


where $\tau_{aer}$ is the AOD, m is the optical air mass and $\tau_{rt}$ is the attenuation due to Rayleigh scattering and other trace gases for cloudless conditions.

Using Equation 1, we conclude that an error of 1% in $V_o(\lambda)$ results in an AOD of 0.01 for an air mass equal to 1. The WMO-GAW specifications call for traceability requiring 95% uncertainty (U95) within ±0.005 + 0.01/m optical depths, where the

5 first term (0.005) is linked to instrument uncertainties (signal linearity, sun pointing, temperature effects, processing, etc.) and the second term to a calibration uncertainty of 1%.

The WORCC standard group of three PFRs (defined as the "PFR triad") was established in 2005 by WORCC in order to fulfil the WMO mandate on: "homogenization of global AOD through provision of traceability to the World Standard Group (WSG) of spectral radiometers for contributing networks at co-located sites and/or periodic international filter radiometer

10 comparisons, and further standardization of evaluation algorithms." Since 2005, five different well-maintained instruments have been used as part of the PFR triad. Figure 1 shows the long term (12 years) comparison of the PFR triad instruments.

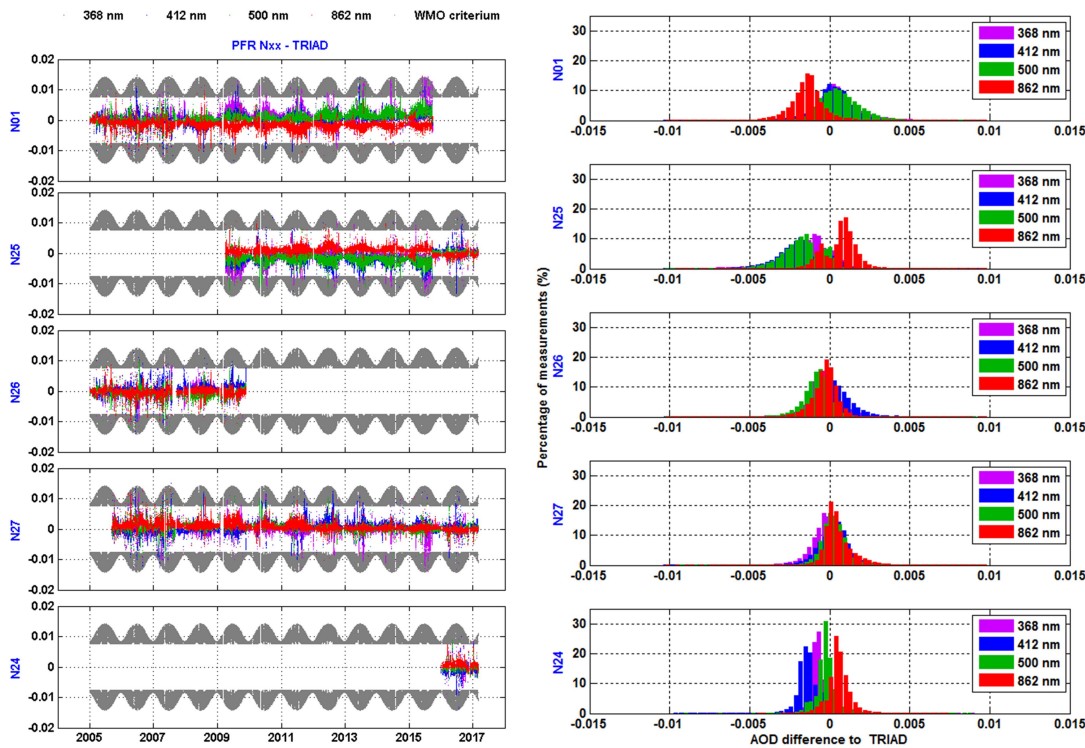

Figure 1: Left panel: difference of each PFR with the mean of the PFR triad at the four PFR wavelengths. Grey areas represent the WMO – U95 limits. Right panel: frequency distribution of these differences for the four measuring wavelengths.

The long term relative stability of each of the five PFRs that were part of the triad is presented in Figure 1. The left panel shows the 1 minute AOD PFR differences, compared using the WMO-U95 criterion at all four PFR measuring wavelengths. It should be noted that all instruments are measuring at WORCC in Davos, Switzerland. They are mounted on the same solar tracking system and their signal is processed using a common processing algorithm. In the 12 years of 1-minute

20 measurement data, more than 99% of retrieved AOD lies within the U95-WMO criterion, at all wavelengths. The right panel of Figure 1 shows in more detail the individual instrument comparisons with the mean triad AOD. As shown, all differences are well within ±0.005 with small shifts for different PFRs and particular wavelengths.


In order to continuously check and maintain the triad stability we have defined a calibration protocol including a frequent visit of instruments performing Langley calibrations at high altitude stations. For this (Langley) calibration method (Holben et al., 1998, Michalsky et al., 2001) the main requirement is the stability of AOD during the measurement Langley periods (half days). Theoretically, this can be achieved anywhere, but in practice AOD is variable during the day, so the current

practice is to perform such measurements at high altitude locations where AOD is very low, thus its variability is very small on an absolute level. Since 2003, Mauna Loa, Hawai, USA (MLO, 19.5°N, 155.6°W, 3397 m a.s.l.), Izana, Tenerife, Spain (IZO, 28.3°N, 16.5°W, 2370 m a.s.l.), and Jungfraujoch, Switzerland, (JFJ, 3580 m asl 46.5°N, 7.9°E), have mainly been used for such Langley calibrations. PFR instruments have been permanently deployed at these stations for certain periods since 2003 and approximately every six months, one of these instruments is returned to WORCC to perform synchronous

measurements in parallel with the triad. Table 1 lists the details of these visits. It describes for each period, the current status of the PFR instruments of the triad, the transferred instrument performing the Langley plots, and the Langley plot measurement location and period.

Table 1. Details of PFR triad Langley calibration measurements.

| Check Year | Triad | | | Comparison Ref | Calibration Type | Comparison Period | |
|---|---|---|---|---|---|---|---|
| **2003** | N01 | N26 | | **N26** | MLO-Langley | 01 Mar. 2000 | 31 May 2003 |
| **2005** | N01 | N26 | N27 | **N27** | MLO-Langley | 01 Sep. 2005 | 31 Dec. 2005 |
| **2009** | N01 | N26 | N27 | **N25** | IZO-Langley | 01 Apr. 2009 | 31 Jun. 2009 |
| **2010** | N01 | N25 | N27 | **N24** | JFJ-Langley | 01 Jan. 2010 | 31 Jan. 2010 |
| | N01 | N25 | N27 | **N22** | MLO-Langley | 01 Jun. 2010 | 31 Jun. 2010 |
| **2011** | N01 | N25 | N27 | **mean of N01,N25,N27** | | | |
| **2012** | N01 | N25 | N27 | **N21** | IZO-Langley | 01 Oct. 2012 | 31 Dec. 2012 |
| **2013** | N01 | N25 | N27 | **N06** | IZO-Langley | 01 Aug. 2013 | 31 Aug. 2013 |
| **2014** | N01 | N25 | N27 | **mean of N01,N25,N27** | | | |
| **2015** | N01 | N25 | N27 | **N06** | IZO-Langley | 21 Sep. 2015 | 28 Sep. 2015 |
| | N01 | N25 | N27 | **N21** | IZO-Langley | 21 Sep. 2015 | 28 Sep. 2015 |
| | N01 | N25 | N27 | **N24** | MLO-Langley | 21 Sep. 2015 | 28 Sep. 2015 |
| **2016** | N24 | N25 | N27 | **N06** | IZO-Langley | 01 Oct. 2016 | 31 Dec. 2016 |
| **2017** | N24 | N25 | N27 | **N21** | IZO-Langley | 17 Mar. 2017 | 14 Apr. 2017 |

The determination of Vo with the Langley calibration method using a 6-month period of measurements requires high accuracy and quantification of the introduced uncertainties. Using a defined calibration method, the Vo accuracy can be traced back to the variability of the Vo determination and is related to the instrument precision and the procedures. Practically, the long term stability of Vo is mainly related to degradation/changes in the transmission of the optical

interference filters, or hardware related failures/changes that are linked to changes in the instrument signal.



WORCC Langley algorithms use half-days to determine Vo values. The main requirements to accept a half-day Langley determination of Vo are: The AOD stability, the signal stability and the statistics of the retrieved signal versus air mass linear regression using specific air mass limitations. An example of accepted Langley (half-day) measurements for a 6-month period at MLO is shown in Figure 2.

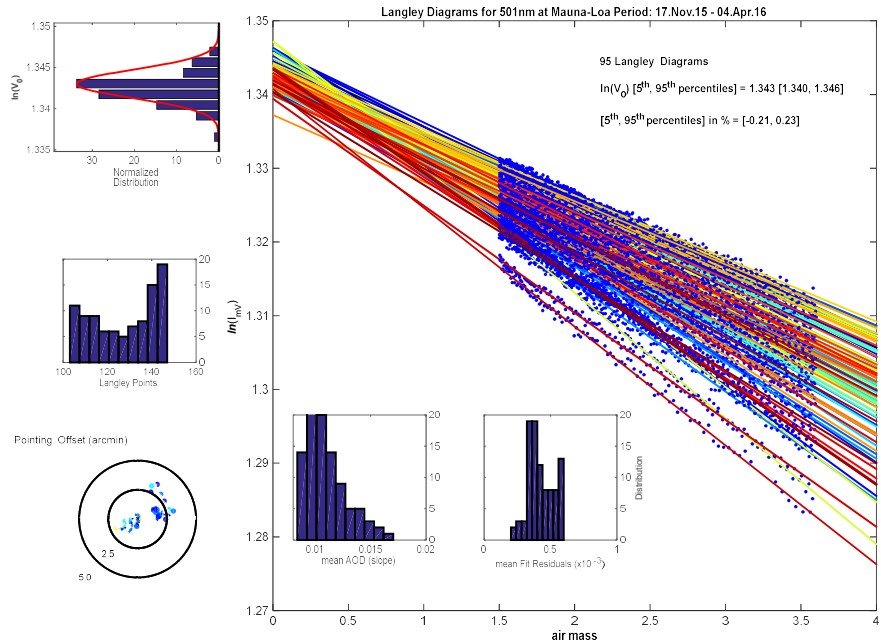

Figure 2. Langley plots for Mauna Loa observatory from November 2015 to April 2016.

Figure 2 shows 95 Langley diagrams/days that have been used to analyse Langley calibration results and related uncertainties. The mean $ln(V_o)$ calculated for this period at 500 nm was 1.343, the standard deviation was 0.002 and the 5th
10   and 95th percentiles were 1.340 and 1.346, respectively. The distribution of $ln(V_o)$ values is also shown together with statistics for mean AOD values (0.010 - 0.015 at 500 nm). The distribution and the normal distribution are shown in the upper left sub-plot of Figure 2.

Based on Eq. 1 the AOD uncertainty, $\delta AOD_{V_o}$, related to the Langley calibration factor equals $\frac{\delta ln(Vo)}{m}$ where $\delta ln(V_o)$ is the
15   uncertainty in $ln(V_o)$. The uncertainty of $ln(V_o)$ can be described by the coefficient of variation (standard deviation / mean, (CV)) or in the case of a normal distribution by the standard error (standard deviation divided by the square root of the number of measurements, (SE)). For the particular example in Figure 2, the calibration uncertainties are shown in Table 2.

Table 2. Calibration uncertainties derived from 6 months of Langley calibration measurements shown in Figure 2.

| PFR wavelength (nm) | Number of Langley plots (N) | Mean ($ln(V_o)$) | Standard deviation | $\delta ln(V_o)$ (CV) $m=1$ | $\delta ln(V_o)$ (SE) (norm. distr.) $m=1$ |
|---|---|---|---|---|---|
| 368 | 75 | 1.438 | 0.002 | 0.0013 | 0.00015 |
| 412 | 93 | 1.308 | 0.002 | 0.0015 | 0.00015 |
| 500 | 95 | 1.343 | 0.002 | 0.0014 | 0.00014 |





| 863 | 56 | 1.276 | 0.001 | 0.0007 | 0.00009 |
|---|---|---|---|---|---|

As described above, this uncertainty is directly related to the calculated $\delta AOD_{Vo}$ uncertainty and is equal to $\delta ln(Vo)$ when m = 1. In Figure 3, $V_o$ values at 500 nm and 865 nm are shown as a function of time for measurements of the PFR instrument N06 that has measured at Davos from 2000-2005 and at IZO from 2005 to 2016. In addition, the evolution of the SE is
shown.

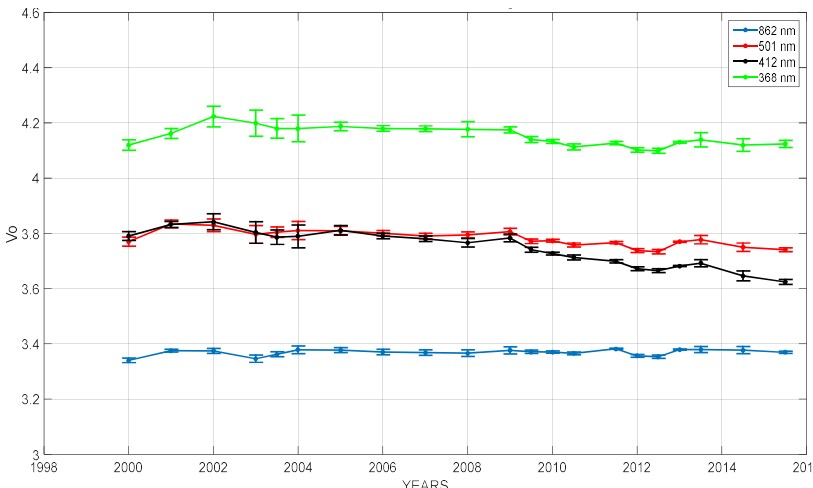

Figure 3. Long-term $V_o$ values for PFR-N06 while at Davos and IZO for the four PFR channels. Bars indicated the standard error.

Each of the data points in Figure 3 represents average $V_o$ values at the end of the averaging period which varies between 3 and 6 months. PFR N06 has exhibited good stability since 2000. All instrument filters have not changed more than 0.1 in $V_o$ units which corresponds to maximum changes in AOD of ~0.02 when m = 1. However, the 412 nm filter is an exception which has apparently degraded since 2009, where the maximum changes in AOD (at m = 1) have been ~0.05 from 2009 to
2016. It has to be noted that all the above described changes have been taken into account in order to calculate the corresponding AOD for the individual periods. Results in Figure 3 illustrate the stability of PFR instruments over time. The very low filter response changes over long term periods increases the statistical validity of each six month Langley calibration period.

### 3 GAW-PFR Network

A primary task of WORCC is the implementation of a global trial network at selected GAW stations with the objective of demonstrating that PFR instruments together with standard calibration techniques and quality assurance procedures can be used to determine AOD with a precision adequate for the fulfilment of the objectives of GAW (WMO, 2001). In addition, it is intended that long term high resolution AOD measurements are conducted and analysed at selected GAW locations.
The locations together with their characteristics in terms of aerosol sources and their period of measurements are described
in Table 3. Bratts Lake and Mace Head measurements were unfortunately discontinued in 2012 and 2015, respectively, due to logistical aspects. However, Valentia (Ireland), Troll (Antarctica), and Marambio (Argentina) have since been added to the core of GAW-PFR stations.



Table 3. GAW-PFR station details, location characteristics and AOD time-series information.

| Station (abbreviation) | Lat. | Lon. | Altitude (m) | Country | Type of Location | Main Types of Air-Masses | PFR AOD Time-Series | Previous studies |
|---|---|---|---|---|---|---|---|---|
| Alice Springs (ASP) | 23.80°S | 133.87°E | 547 | Australia | desert | remote continental | 2002 – present | *Mitchell et al. 2017* |
| Bratts Lake (BRA) | 50.28°N | 104.70°W | 576 | Canada | prairie, agricultural | remote continental | 2001 – 2012 | *McArthur et al. 2003* |
| Danum Valley (MAL) | 4.98°N | 117.84°E | 436 | Malaysia | tropical forest | remote continental | 2007-2016 | |
| Hohenpeissenberg (HPB) | 47.80°N | 11.02°E | 995 | Germany | pre-alpine, rural | rural | 1999 – present | *Ruckstuhl et al., 2008; Nyeki et al. 2012* |
| Izana (IZO) | 28.31°N | 16.50°E | 2371 | Spain | island | free-troposphere | 2001 – present | *Barreto et al. 2014* |
| Jungfraujoch (JFJ) | 46.55°N | 7.99°N | 3580 | Switzerland | high-alpine | free-troposphere | 1999 – present | *Ruckstuhl et al. 2008; Nyeki et al. 2012* |
| Mauna Loa (MLO) | 19.53°N | 155.58°W | 3397 | USA | island | free-troposphere | 2000 – present | *Dutton et al. 1994* |
| Mace Head (MHD) | 53.33°N | 9.89°E | 20 | Ireland | coast | marine boundary layer | 2000 – 2015 | *Mulcahy et al. 2009* |
| Ny Ålesund (NYA) | 78.91°N | 11.88°N | 17 | Svalbard | Arctic coast, island | Arctic/marine boundary layer | 2002 – present | *Herber et al. 2002* |
| Ryori (RYO) | 39.03°N | 141.83°E | 230 | Japan | coast | marine boundary layer | 2002 – present | |

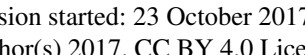



| Cape Point (CPT) | 34.35°S | 18.49°E | 230 | S. Africa | coast | marine boundary layer | 2007-present | *Nyeki et al. 2015* |
| Mt. Waliguan (WLG) | 36.28°N | 100.90°E | 3810 | China | high-mountain | free-troposphere | 2007-present | *Che et al. 2011* |
| Valentia (VAL) | 51.94°N, | 10.24°W | 24 | Ireland | coast | marine boundary layer | 2007 - present | |
| Marambio (MAR) | 64.24°S | 56.62°W | 205 | Argentina | coast | marine boundary layer | 2005 - present | *Tomasi et al. 2015* |
| Troll (TRO) | 72.01°S, | 2.54°E | 1309 | Antartica | polar | free-troposphere | 2012-present | *Tomasi et al. 2015* |

In addition to the core GAW-PFR instruments, 30 other locations exist that are performing AOD measurements using PFR instruments belonging to individual users/institutes. An overview of data flow and availability for every location is shown in Figure 4.

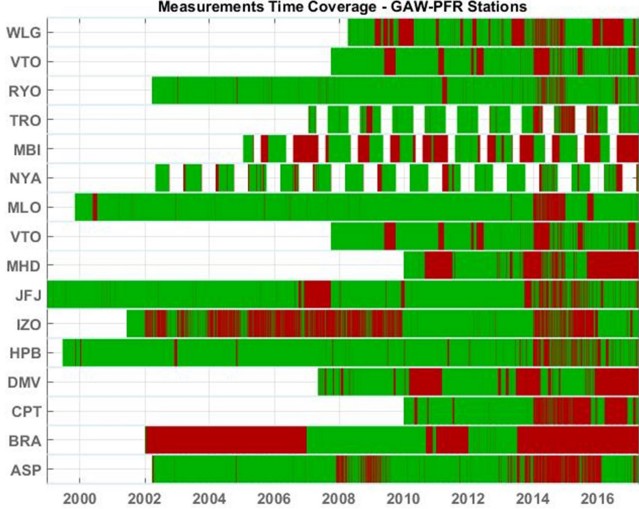

5    Figure 4. Data coverage of GAW-PFR stations. Green indicates periods with data availability and red with missing data.

At high latitude locations such as Troll and Ny Ålesund, AOD measurements can only be performed during part of the year due to the luck of direct sun. Big gaps (red colors) are linked with instrument damage (e.g. MHD, DMV and CPT) due to various reasons (corrosion, lightning etc.).

10    Smaller (red) gaps are due to instrument recalibrations through transfer and measuring in parallel with the PFR triad at Davos, Switzerland.



### 3.1. Instrument calibrations

Instruments are regularly calibrated (every one to two years, depending on instrument related and logistical aspects). The calibration of the filter radiometer has to be assured with an uncertainty of ±1% in order to achieve the required AOD

uncertainty to be within the U95-WMO limits. Quality assured AOD data can only be obtained when pre- and post-deployment calibration constants are available. That means that AOD data for a certain location and period can only be considered as final after the recalibration of the instrument which is performed at the end of the specific period.
 Post-calibrations can be obtained by different methods:

- WORCC calibration certificate

Is obtained by instruments/stations that have their PFR calibrated at Davos, against the WORCC triad. Polar and high altitude stations are often sent for calibration annually during polar night/longest night periods. For other stations, a re-calibration should be performed every 12 to 24 months. This method implies that preparation of the final version of the AOD data might be postponed by 1 to 2 years. Comparison of each of the instruments with the triad is performed based on the

WMO criteria for AOD inter-comparisons. The inter-comparison lasts at least five cloudless days.

The extraterrestrial calibration constants of an instrument x, $V_{0xR}$ are determined by using 1-minute measurements $S_{ix}$ of the instrument to be calibrated, and synchronous measurement $S_{iR}$ of the reference (mean of the three triad PFRs) instrument.

$$V_{0xR} = \frac{1}{N} V_{0R} \sum_{n=1}^{N} \frac{S_{ix}}{S_{iR}} \qquad \text{(Eq. 2)}$$


The daily calibration constants $V_{0xR}$ according to (Eq. 2) are determined for days where a number (N ≥120) of solar measurements unobstructed by clouds were collected. Comparing the instrument measurement signal to that of the triad, a new $V_{0xR}$ is calculated and compared with the last one used. Details of such an analysis for a single day are shown in Figure 5. Instrument signals are shown in Figure 5a, then the percentage differences of 1-minute data are calculated for all four

wavelengths in Figure 5b. An example of the calculated AOD differences from the triad before and after the calibration is finally shown in Figure 5c. For the specific instrument, the $V_{0xR}$ differences for the particular day were up to 0.5% and depended on the wavelength. The impact of this difference in AOD calculation is an air mass dependent difference of 0.008 to 0.003.

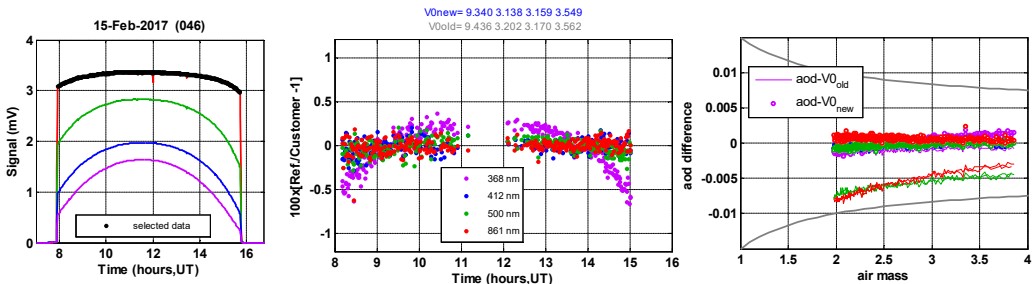

Figure 5. Calibration of an instrument against the triad: a. Measurement signals at four wavelengths, b. comparison of instrument signals, c. Differences of instrument vs triad retrieved AOD using the old and the new calibration $V_{0xR}$

The average values $\overline{V}_{0xR}$ and standard deviations $\sigma_{0x}$ of the daily mean calibration constants from each reference instrument are averaged to give the final calibration constants $V_{0x}$ with an expanded uncertainty U95:



$$V0_{U95} = 1.96 \sqrt{\sum_{i=1}^{N_{ref}} \left( \frac{\bar{V}_{0xR} - V_{0x}}{2\sqrt{N_{ref}}} \right)^2 + \sum_{i=1}^{N_{ref}} \left( \frac{\sigma_{0xR}}{\sqrt{N_{days}}} \right)} \qquad (3)$$

The two terms under the square root in equation 3 describe the combined statistical (comparison) and triad uncertainty
during the calibration period.

For a normal distribution, $V0_{U95}$ corresponds to a coverage probability of approximately 95%.

The calibration is considered successful when the coefficient of variation CV = $V0_{U95}$ / $V_{0x}$ becomes smaller than ±0.5% for
all 4 channels of the instrument to be calibrated. This limit is typically reached after 3 to 5 days of comparison.  In Figure 6,
we show $V0_{U95}$ calculated over an extended calibration period of 35 days for a PFR instrument measuring against the triad.
For each of the four PFR wavelengths, $2\sigma$ within the period is on the order of 0.4% to 0.9% of the mean Vo. In addition, we
have calculated the $V0_{U95}$ limit in percent using Equation 3 for the instrument under calibration with each of the three PFRs
that are part of the WORCC triad.  For this particular case, all wavelengths are within the limit CV < 0.4% after three days
and after 10 days within CV < 0.2%.

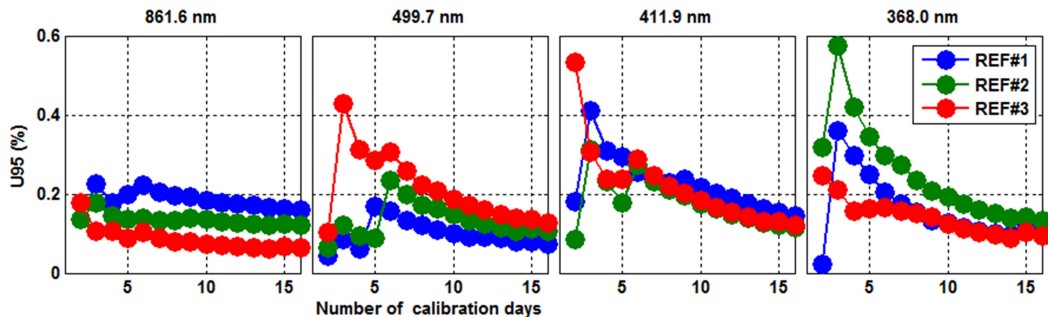

Figure 6. Graphs of $V0_{U95}$ calculated over an extended calibration period of 35 days for a PFR instrument measuring against
each of the three triad instruments for the four PFR wavelengths.

• Langley sites


Calibrations can be obtained by statistical analysis of objective Langley plots collected in-situ over an extended period of
time at high altitude (IZO, JFJ, MLO) or remote background (ASP, BRA, TRO) sites. Such an evaluation of Langley plots is
routinely performed every six months using Langley results from six months before and after the anchor dates of 1 January
and 1 July for each year. This method implies that annual quality assured data become possible in July or August of the
following year.

A smaller (<1%) calibration uncertainty can be expected and is required for Langley sites where AOD is lower than
elsewhere and inconsistent calibration may lead to erroneous conditions such as an inverted Ångström relation (channels
crossing over) or negative AOD.

Calibration $V_o$ values calculated for the four AOD channels are used in order to retrieve the AOD. For the data obtained
between two calibrations a calibration slope of $V_o$ values is applied. If the differences between two calibrations are larger
than 2% then an in-situ estimate of the instrument stability is investigated from a number of in-situ Langley plots or cross-
calibrations between different PFR channels.  This is conducted in order to determine non-linear (over time) changes (steps)
of instrumental $V_o$ values.

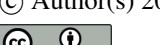


### 3.2 Quality control

After finalizing the calibration constants to be used for the AOD retrieval, a series of QA/QC procedures are used before finalizing the AOD data.

- Check for ancillary data

Ancillary data of atmospheric pressure and total ozone are needed to retrieve the Rayleigh and ozone optical thicknesses, respectively, according to Equation 1. Atmospheric pressure measurements, required for Rayleigh scattering, should be accurate to about 3 hPa. This accuracy is readily achieved by meteorological grade barometers built into new PFR loggers. Accurate pressure data are requested from each station and compared to the daily PFR logger values. If the mean differences

are larger than ~3 hPa, then the atmospheric pressure is corrected and all data are reprocessed. The use of average atmospheric pressure data, over a day or longer periods, can lead to wavelength dependent AOD retrieval errors and to large Ångström exponent errors.

Total column ozone values are needed to correct optical depth at 500 nm for ozone absorption. As the absorption coefficient at 500 nm is low, total ozone needs to be known to ±30 Dobson units, or 10% of typical values, for an uncertainty of ±0.001

optical depths at 500 nm. GAW-PFR uses (AURA) satellite overpass observations by the ozone monitoring instrument (OMI) for daily operations (McPeters et al., 2015). OMI values are validated to in-situ observations for stations operating a Dobson or Brewer instrument. Where available, total column ozone may be found at the World Ozone and Ultraviolet Radiation Data Centre database (www.woudc.org). Figure 7 shows the evolution of daily mean pressure used for the Rayleigh calculations and ozone values at Davos, Switzerland as measured in situ with a Brewer spectrophotometer, and

from the OMI ozone retrieval.

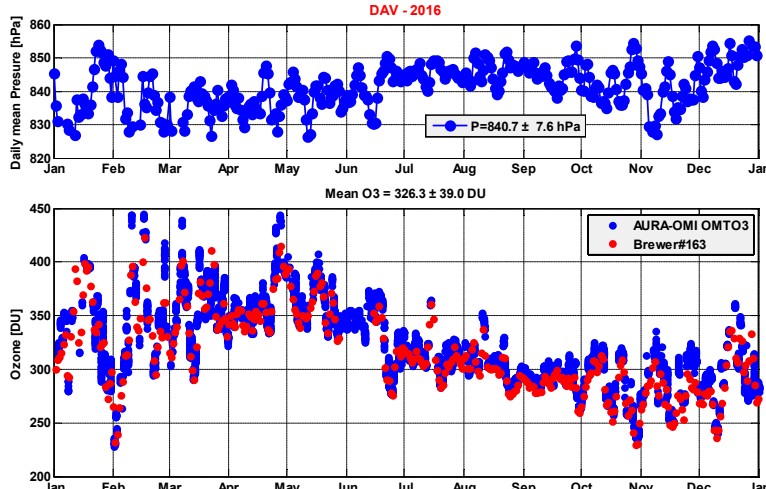

Figure 7. Upper panel: Daily mean pressure during 2016 at Davos, Switzerland. Lower panel: Brewer and OMI ozone values.

- Corrections for temperature, dark signal

The PFR sensor temperature is checked for deviations from its active stabilized set-point, indicating potential problems during extremely hot or cold ambient conditions. The PFR dark signal is checked for values >0.25 mV, and if found on approximately 5% of days, a correction is applied. The dark signal is the mean signal when the solar elevation is less than -

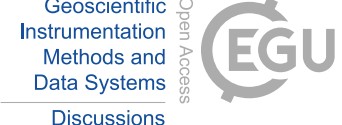



6°, i.e. below the horizon. The temperature dependence of its PFR is based on characterization measurements in a climate chamber. Corrections are applied only in cases when the dependence on Vo is more than ±2% for the range from -20° to 40°C.

• Sun Pointing

In order to ensure that the full solar disk is included in the field-of-view (FOV) of the instrument an accurate sun-tracking system is required. While a PFR instrument can be readily aligned to the sun with the required accuracy, a solar pointing monitor of the PFR is included in order to control the sun-pointing accuracy. This monitor consists of a four quadrant silicon

detector that is illuminated through a pinhole of 1 mm diameter at a distance of 70 mm. When the light is centered, all four quadrants produce equal signals. By subtracting signals from paired pixels, the sun spot can be localized (Wehrli, 2008).
Figure 8 shows four examples of perfect to bad instrument pointing on four cloudless, low aerosol concentration (AOD < 0.1) days. The instrument pointing is shown in the upper plots. The first (from the left) case shows a perfect pointing accuracy where 328 measurements during the day are almost identical in terms of pointing direction. The second case shows

when all measurements can be found inside the 15 arcmin limit. Finally, the last two cases show instruments with pointing issues where only 62.9% and 59.5% cases respectively are inside the 15 arcmin limit, where in the last case a number of measurements are outside the 25 and 30 arcmin limit.
The result on the measured signal (which has a direct impact on calculated AOD) is shown in the lower panel, where the first two days show a very smooth daily pattern while artificial signal features can be seen in the last two cases. The use of these

data will end up in an artificially overestimated AOD retrieval.

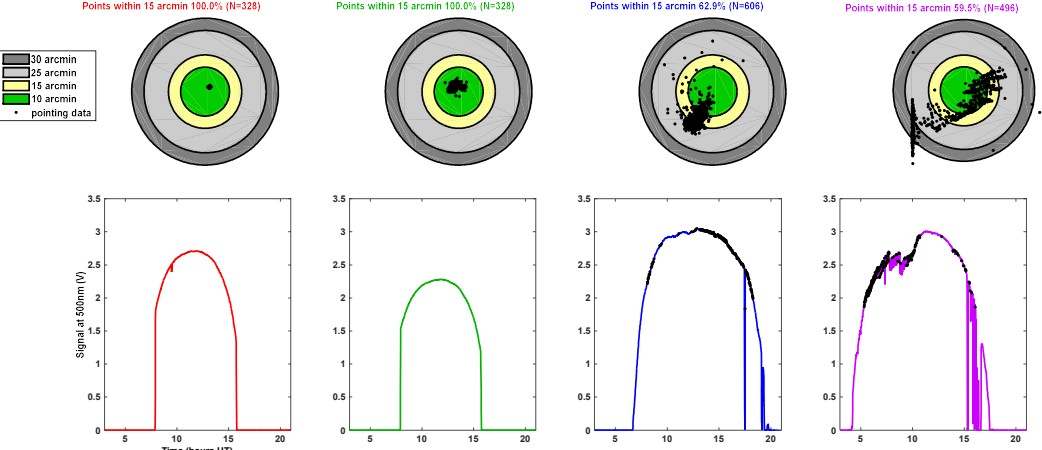

Figure 8. Example of good and bad pointing. Upper panel: instrument pointing, lower panel: instrument signal at 500 nm.
The two lower right panels include some cloud related signal changes after 15 hour U.T.

• Check for crossing of wavelengths
An additional quality control check detects instances when AOD at one of the four PFR wavelengths is less than that for a higher wavelength. This quality check is mainly performed in order to detect the erroneous performance of one of the four

channels. The test also becomes very important when low AOD values are measured, which is the case at a number of GAW-PFR stations. Small errors related to the calibration of one channel can be easily identified, as this results in a

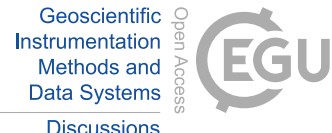

wavelength crossing of AOD. During evaluation of the data, the processing software tools give the opportunity to define the limits of the accepted offset for the wavelength crossing.

An example and the graphic representation of wavelength crossing is shown in Figure 9. Based on the fact that AOD at lower wavelengths has to be at least equal or higher than that at higher wavelengths, colored points show the correlation of

pairs of AOD values at different wavelengths and black points represent the cases when wavelength crossing occurs. The figure is composed of 70350 1-minute, cloudless measurements from the year 2015 as recorded at Cape Point, South Africa. For example, as many as 14.3% of data in the 412 – 500 nm panel (red-blue section) do not pass the wavelength crossing test, so they are discarded form the post processing analysis.

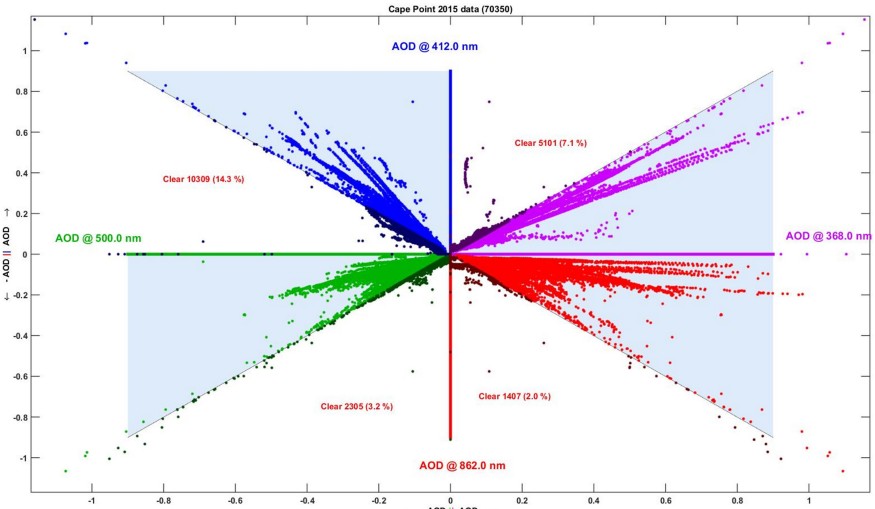

Figure 9. Correlation curves of the four PFR measuring wavelengths. Colored points represent the data that have passed the wavelength crossing test and black ones those that have not.

- Other issues

The spectral bandpass of all PFR instruments has been characterized for their effective spectral wavelength and bandwidth.

This is determined as the average wavelength weighted by the spectral response and equivalent width of a rectangular bandpass with equal throughput as the filter. Minimum and maximum central wavelengths (and bandwidths) that have been calculated were: 367.2-367.7 (3.5-3.7), 411.8-412.6 (4.3-4.4), 500.6-501.5 (5-5.1) and 861.3-863 (5.5-5.6). These measurements were performed using illumination by a grating monochromator (Jobin Yvon HR640) with 0.6 nm spectral resolution. Lately, a pulsed tuneable laser system for the characterisation of spectrometers and filter radiometers has been

available at PMOD/WRC. Test measurements with PFR instruments did not show significant differences to the older characterization measurements.

### 3.3 Cloud flagging

As AOD measurements cannot be performed under cloudy conditions, a cloud detection algorithm is used for the PFR

measurements. Three different criteria are used (Wehrli, 2008):

a. The instrument signal derivative with respect to air mass is always negative. The method has been developed and described in detail by Harrison and Michalsky (1994). For cases when air mass values < 2 and the influence of clouds on the



noon-side of perturbations cannot be easily detected, we compare the derivative with the estimate of the clear Rayleigh atmosphere and flag it as cloudy if the rate of change is twice as much (objective method).

b. The use of a test for optically 'thick' clouds with AOD >2.

c. The use of the Smirnov triplet measurement (Smirnov et al., 2000) by calculating AOD and looking at the signal

variability for three consecutive minutes (triplet method).

An example of the use of these three criteria can be seen in Figure 10, where a day with variable cloudiness at Davos is presented.

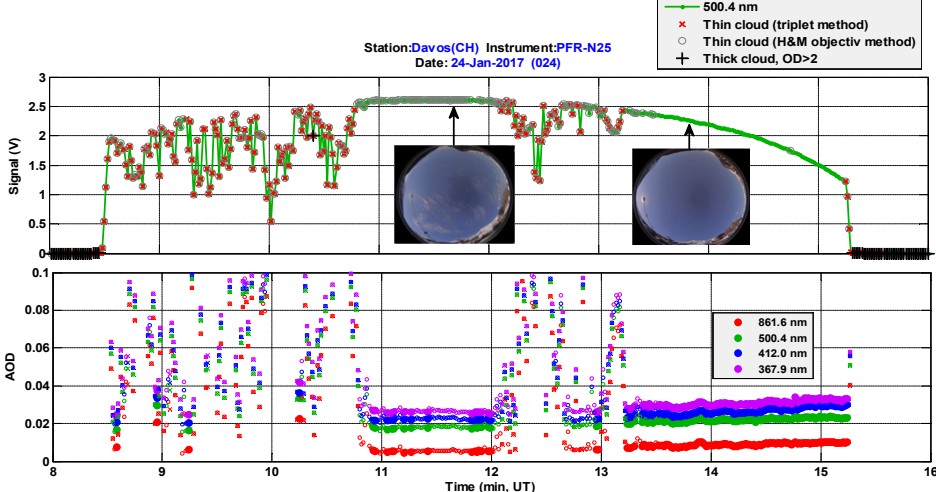

Figure 10. Example of a day with variable cloudiness, Upper panel: instrument signal at 500 nm and minute-by-minute application of the three cloud flagging methods. The two inset pictures show a 360° view of the sky using a cloud-camera. Lower panel: calculation of AOD at four wavelengths.

For this particular day, all three criteria are applied. In the early morning and evening, the thick cloud criteria are applied.

Then both the triplet and the objective method are applied due to variable cloudiness in front of the sun. However, there are times during the day when only the objective method is applied (thin clouds in front of the sun as seen in the first picture that is superimposed in Figure 10). During the last part of the day (second picture), clouds completely disappear and cloud flagging is set to zero which means that all three criteria are passed. It has to be noted that cloud flagging is always kept as a constant number describing which one of the three criteria, or which combination is valid at a certain minute.

The lower panel shows the calculation of AOD for the whole day, with obvious deviations due to cloud occurrence for the parts of the day when both criteria are fulfilled. It is interesting to see the 10:50 to 12:00 period which is a difficult period when defining the presence of clouds only with direct sun measurements. For this particular period, even if the AOD is low, the objective method shows the presence of thin clouds in front of the sun.

## 4 Final AOD data

During the calibration and quality control procedure, three levels of data are defined.

Level 1: These are the raw signal data as measured by the PFR instrument at the four different channels.

Level 2: These are AOD values. The data are produced at each measuring station using standardized software including: QC tests, cloud screening, sun-tracking details and signal-to-AOD conversion using an existing calibration file. Each of the

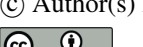



mentioned test results is characterized by a specific flag. In addition, the true solar elevation is calculated and included. None of the level 1 data is discarded.

Level 3: AOD data are re-evaluated at WORCC which include AOD results and Ångström coefficients. Additional checks are included such as the detection of wavelength crossing $AOD(\lambda_1) > AOD(\lambda_1)$, where $\lambda_1 > \lambda_2$. In addition, a day-to-day

visual inspection is performed in order to identify other technical issues or the possible presence of undetected clouds. For the latter, additional cloud flags are included in the final data files. Data control of level 3 data includes overviews of the instrument's tracking performance, wavelength crossing, and ancillary data.

Hourly data records are prepared from quality assured level 3 data which are then submitted to the World Data Center for Aerosols (WDCA) hosted by the Norwegian Norsk Institutt for Luftforskning (NILU; ebas.nilu.no). Final data files include

the mean, median, standard deviation and the number of 1-minute samples used to calculate the hourly value at all four wavelengths.

In order to calculate hourly, daily and monthly statistics, we apply the following criteria:

- A minimum of 50 cloudless 1-minute measurements per day are required to calculate daily statistics. In this case, we eliminate days with less than one hour of sunshine.

- A minimum of six 1-minute cloud free measurements are required to calculate the hourly mean.

- A minimum of 30 hourly values are required to calculate the monthly mean.

- Measurements that lie beyond two standard deviations for an hourly mean are considered outliers, as they are considered to be affected by cloud contamination.

Monthly statistics can be presented with different approaches. In most studies, AOD is usually reported as the arithmetic mean and associated standard deviations over a selected period. This is based on the hypothesis of an underlying normal distribution. However, AOD is often better characterized by a lognormal distribution and described by geometric mean and standard deviation. Based on a statistical analysis of skewness and kurtosis in a multi-year and multi-station AOD data set, O'Neill and co-authors (O'Neill et al., 2000) have shown that a lognormal distribution systematically provides a more robust

base for reporting AOD statistics than the normal distribution. Using long term series of the final-selected 1-minute AOD data, users can then try to draw conclusions on the AOD climatology of each station, the aerosol changes if any, or the daily monthly and annual patterns. As an example of the three Langley calibration related stations IZO, DAV and MLO, monthly means calculated from 15 years of measurements are presented in Figure 11. IZO and MLO are the Langley calibration sites and DAV the triad host site.

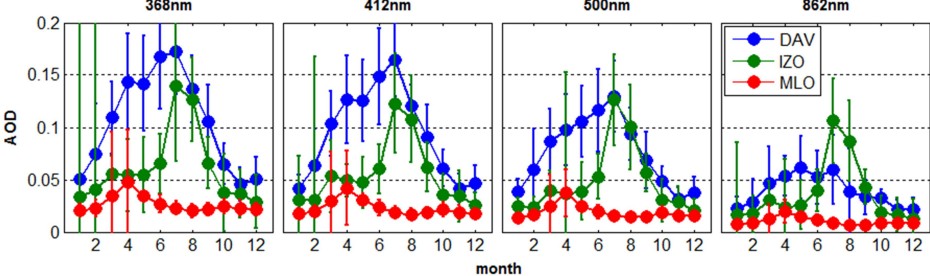


Figure 11. Monthly means and 1σ standard deviations for Davos, Izaña and Mauna Loa using 15 years of 1-minute quality controlled PFR measurements at four wavelengths.





For the particular sites that are all considered to have low AOD, we can clearly see that Davos shows an increase in AOD during the summer months, while the other two sites show much lower AOD with the exception of Saharan dust intrusions at Izaña for the July-September months.

When comparing MLO and IZO statistics, we calculate long term $AOD_{550}$ means of 0.050 and 0.020, respectively, and geometrical means of 0.033 and 0.017. Respective geometrical standard deviations are about 1.7 for MLO and 2.6 for IZO meaning that AOD varies from 60% to 170% of 0.017 for MLO and from 38% to 260% of 0.033 for IZO. This is linked with the Saharan dust events (AOD outliers for a normal distribution) that affect IZO.  An overview of the GAW-PFR AOD time series at all stations will be reported in a future study (Kazadzis et al., paper in preparation).

## 5 Summary and Conclusions

"AOD is the single most comprehensive variable to assess the total aerosol load of the atmosphere and represents the least common denominator by which ground based remote sensing, satellite retrievals and global modelling of aerosol properties are compared" (WMO, 2016). According to the WMO, multi-wavelength AOD is one of the essential variables that critically contribute to the characterization of Earth's climate. In addition, the Global Climate Observing System (GCOS) also includes atmospheric aerosols including AOD as an essential climate variable. Finally, the European Space Agency has included aerosols and AOD as one of the 10 climate change initiative (CCI) variables to be investigated with a view towards building space-based databases.

In order to monitor AOD over the long-term and provide data of traceable quality, the World Optical depth Research and Calibration Centre (WORCC), Davos, was established by the WMO Global Atmosphere Watch (GAW) program. Fifteen existing GAW baseline stations were chosen for the deployment of PFRs (Precision Filter Radiometer; in-house manufacture). Quality controlled and assured AOD data from this GAW-PFR network (www.pmodwrc.ch/worcc) are being submitted by WORCC to the World Data Centre for Aerosols (ebas.nilu.no).

Under conditions of low aerosol loading, e.g. AOD < 0.1 at 500 nm, a calibration error of 1% results in an error of ~12% in the mean daily AOD. WMO has recommended (WMO, 1994) an absolute limit to the estimated uncertainty of 0.02 optical depths for acceptable data and <0.01 as a goal to be achieved in the near future. These specifications require a calibration uncertainty better than 2% to be achieved for spectral radiometers. In addition, measurement quality control and quality assurance in different processing levels of the actual measured direct sun signals or retrieved AOD have to be included.

The calibration hierarchy of any network of sun-photometers is linked with the instrument performance and stability over time. Instruments which do not exhibit good stability (e.g. filter degradation) over time, tend to utilise short periods for Langley calibrations where the instrument response can be considered constant. This can impact the calibration constant uncertainty through the limited number of measurements and the statistical analysis that is used.  The PFR development and construction has been based on the use of specific hardware and manufacturing techniques that make them reliable for long term measurements without rapid interference filter changes (e.g. Fig. 2). This provides the opportunity of using longer periods for collecting Langley calibration results and thus results in better statistics for the determination of the calibration constants.

Quality control of routine WORCC/PFR measurements includes a number of measurement related checks, including: the optical window cleanliness and the accuracy of the sun-pointing. In addition, a number of parameters such as pressure, ozone and $NO_2$ concentrations have to be measured/assumed/modelled.  Further QC procedures involve data evaluation; especially





rejecting measurements with wavelength related drifts (crossing) and suspected cloud contamination in the line-of-sight. Cloud screening becomes a difficult task especially in the case of optically thin clouds that cannot be easily distinguished from AOD associated with coarse mode aerosols. Finally, quality assurance od AOD data mainly includes the determination of a proper calibration (extraterrestrial signals) within the required uncertainty.

WORCC has defined a protocol for calibrating the PFR instruments by maintaining a triad of reference PFRs that exhibit differences well within (more than 99% of 1-minute data over a 12 year period) the U95 WMO criterion. The procedure includes systematic checks including comparisons with instruments that perform measurements (Langley calibrations) at high altitude stations.

One of the aims of WORCC is the provision of instrumentation and protocols for uniform global measurement and records of AOD and the maintenance of the radiometric reference for such measurements. So in addition to the hosting and maintenance of the AOD triad, WORCC hosts the filter radiometer comparison every five years (e.g. WMO, 2016) and maintains long term AOD measurements at the main calibration sites of other aerosol networks such as AERONET (Mauna

Loa, USA; Izaña, Spain) and SKYNET (Chiba, Japan; Valencia, Spain).

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
