# Peer review of "The World aerosol Optical depth Research and Calibration Center (WORCC), Quality assurance and quality control of GAW-PFR AOD measurements"

_Geoscientific Instrumentation, Methods and Data Systems, 2017_

## Referee Comment (RC1) · L. Doppler (Referee) · 20 Nov 2017

**1-GENERAL COMMENTS**

This paper presents an institution and its services (the WORCC) contributing to the gain of quality dataset in geoscience/atmosphere composition (aerosol optical depth: AOD). The paper presents also the methods used at WORCC to improve the quality of the measurements (QA/QC about the collected data, calibration of the instruments) thus this paper addresses totally relevant scientific questions within the scope of GI.

[Figure]

The innovative part of the paper is that it is the first paper presenting in detail the activity and methods of this key institution (WORCC) doing a key activity for the QM of GAW-PFR network dataset. Until now, there were only some fractions of descriptions widespread in many chapters of many WMO reports, moreover in a too old literature that could not synthesize the many years of dataset of GAW-PFR and all the lessons of its long experience. This synthesis of experience is well presented in is this paper.

Substantial conclusions are reached: the paper summarizes in a very clear way the methods to apply (and already applied in the institution of reference that is presented: the WORCC) in order to have a well quality of management of a worldwide network of AOD measurements using sun photometry.

The scientific methods used are well described their validity are discussed, a good balanced use of figures and mathematic equations contributes to a clear outline of them. The long dataset and experience of calibration and data flagging presented in this paper contain enough results to develop pertinent interpretations and conclusions.

One aim of the paper is clearly to describe precisely a QM protocol that is destined to be reproduced in other stations or other networks of sun photometry of AOD. The procedures are well described and this aim is in my opinion perfectly reached.

The references list is completed enough giving proper credit to current and past work related to this topic, even if some technical improvements in the way to cite the literature references would be welcome (see below). The number of references is good balanced and the references are of excellent quality. Thanks to this literature work, the authors could clearly put forward their own contribution to the topics approached in this paper (sun photometry, quality management of AOD dataset and worldwide networks, calibration of the instruments of an AOD network).

The title of the paper reflects the content of the paper in a good way; the abstract is a good complement of the title and a concise and truth summary of the paper.

The overall presentation is well structured, and despite some minor details (to which I suggested improvements in the part below named "technical comments") clear expressed.

The language is fluent and precise and it is an obstacle neither to get rapidly a good comprehensive view of this work nor to understand the technical and mathematical details

The mathematical formulae are fairly shown, some improvements would be welcome as I suggest it in "technical comments", especially for some equations a lack of definition of all parameters used in the equations. Nevertheless, all equations are right, without mistake and globally well understandable.

I would suggest some minor improvements to be done: More information should be given in the introduction about the history of sun photometry; the way to cite the references should be more rigorous; and some equations should be better explained (more details, definition of the parameters). These are minor corrections to be done and I tried to help the authors by making concrete suggestions in the parts below "specific comments" and "technical comments".

Despite these minor corrections that I suggest, the article is of very good scientific quality of excellent significance and globally of good presentation. This justifies my evaluation here above and the fact that I suggest the editor to accept the manuscript and to ask for minor corrections.

2 SPECIFIC COMMENTS

Note: X.Y means Page.Line, ex 5.17 is page 5, line 17

2.1 Principal specific comments

- Figure 3: Comparing the uncertainty bars and the slopes, this figure does not show an improvement of the quality of the calibration, when WORCC moved the calibration site from Davos to Izaña. To which is IZO a site of better quality for Langley calibration

than PMOD-WRC?

- Chapter 3.1. – Instrument calibration: In the case of a calibration against the triad. What is the strategy? Do you try to have a representative panel of airmass and AOD or do you prefer to focus on low AOD in order to test the sensitivity for low values (avoid negative values, improve detection of AOD for low aerosol masses)

- About criteria for the statistics:

o For monthly statistics: "A minimum of 30 hourly values is required" -> implies 2 days of measurements are enough if they are full and in summer. Is it reasonable ? Don't you want to introduce a criterion of amount of days per month?

o No criterion of repartition of the minutes during the day. My question about the daily mean: Maybe for one site we have, because of the clouds or the availability of the horizon (mountains), a morning average and in other places an afternoon average -> Would the comparison of the daily AOD at these 2 sites be still pertinent?

2.2 Other Specific comments

- In the introduction (1.21), you cite "AOD has been measured with the use of sun-photometers for more than 50 year (Holben et. al., 1998)". I have two comments to this:

o Holben et al. 2001 (also in references' list) describe better and longer the 50 years long history of AOD measurements with sunphotometers than Holben et al. 1998

o I really suggest you to briefly describe this 50 years story of sunphotometers, and more expansively than Holben et al. did. All the authors of this manuscript are staff members of PMOD-WRC, a very historical institution, this is why, the reader expects from you that you have the ambition and motivation to make this historical description by your own. You can cite Volz (1959, 1969), Flowers, Shaw (1976, 1982), Leiterer and Schulz (wmotd 222, 1988)... And maybe more recent articles describing long time series at specific sites (Weller and Gericke [Met. Zeit. 2005] for MOL-RAO Lindenberg,

Barreto et. al [AMT 2014] for Izana, something about PMOD-WRC, . . .).

- In the introduction (2.17), you mention that "GAW-PFR aims to provide inter-comparison information between networks by overlapping sites". -> Is it only an objective (aim) or are there already studies that make inter-comparison of networks? If there are some studies, please mention them and cite the corresponding publications.

- Chapter 3.2. – Other issues (13.13-21): The QM parameter tested is well described. Could you inform about the threshold of spectral shift that your QC politic allows for the spectral shift of the spectral channel?

- Figure 10, Page 14: It is well shown how each method detects or not some type of clouds. Could you explain what are your own QC using all these different methods? Which data you keep in the Level 2 or Level 3 of GAW-PFR database and which you flag out because you consider them as cloud observations.

3 TECHNICAL COMMENTS

Note: X.Y means Page.Line, ex 5.17 is page 5, line 17

3.1. Citations / references

An effort has to be done concerning the reference citations:

A main comment concerning the citations: When a WMO report is cited, since some reports are very long, please in this case, inform the reader about the chapter(s) where the information can be found. Inform the chapter directly in the reference list (see my suggestions below).

Moreover, you have in your references list, 2 WMO/GAW reports of the year 2016, that you both cite with "WMO, 2016":

- WMO/GAW Report n°227 (Guidelines and recommendations)

- WMO/GAW Report n°231 (FRC-IV)

⇒ I suggest to put in brackets in the references list "WMO, 2016-227" and "WMO, 2016-231", and to cite with "WMO, 2016-227" or "WMO, 2016-227" in the text (see my suggestions below).

For your help, these are all WMO reports citations in the text and after it in my suggestions how you could cite properly:

- 1.19. "WMO, 2016" -> "WMO, 2016-227"

- 1.25. "WMO, 1993" -> keep unchanged, even if I doubt that you do want to cite "WMO, 1995", if not, reference GAW/WMO, report No. 104 (March 1995) can be erased because it would never been cited.

- 2.13. "Wehrli in WMO 2005" -> keep unchanged

- 2.24. "WMO, 2016" -> "WMO, 2016-231"

- 6.22. "WMO, 2001" -> keep unchanged

- 16.12. "WMO, 2016" -> "WMO, 2016-227"

- 16.25. "WMO, 1994" -> It is "WMO, 1993"

- 17.13. "WMO, 2016" -> "WMO, 2016-231"

In the references list, I suggest following changes:

- "GAW Report No. 227, WMO/GAW Aerosol Measurement Procedures, Guidelines and Recommendations, 2nd Edition, WMO- No. 1177, ISBN 978-92-63-11177-7, 2016." -> please list this WMO/GAW report with the same way as the other GAW-WMO reports are cited and specify the chapter of the AOD, and specify that it is named "WMO, 2016-227" in text: "WMO/GAW report No. 227, WMO/GAW Aerosol Measurement Procedures, Guidelines and Recommendations, 2nd Edition, WMO- No. 1177, ISBN 978-92-63-11177-7, (WMO/TD- No. 1177), ISBN 978-92-63-11177-7, August 2016.; Chapter 7. Aerosol Optical Depth (pp. 60 - 67) (in text: WMO, 2016-227)"

- "WMO/GAW The Fourth WMO Filter Radiometer Comparison (FRC-IV), GAW Report No. 231, 2016" -> Please try to use one unique citation style for WMO/GAW reports and specify that it is named "WMO, 2016-231" in text: "WMO/GAW Report No. 231, The Fourth WMO Filter Radiometer Comparison (FRC-IV), November 2016 (in text: WMO. 2016-231)"

- "WMO/GAW report No. 162, Experts Workshop on a Global Surface-based Network for Long Term Observations of Column Aerosol Optical Properties (WMO TD No. 1287), 153 pp, November 2005" -> Specify the chapter: "WMO/GAW report No. 162, Experts Workshop on a Global Surface-based Network for Long Term Observations of Column Aerosol Optical Properties (WMO TD No. 1287), 153 pp, November 2005; Chapter: 'GAWPFR: A Network of Aerosol Optical Depth Observations with Precision Filter Radiometers' (from Christoph Wehrli, pp. 36-39)"

- "WMO/GAW report No. 101, Report of the WMO workshop on the measurement of atmospheric optical depth and turbidity, (WMO TD No. 659), December 1993." -> Specify the chapter: "WMO/GAW report No. 101, Report of the WMO workshop on the measurement of atmospheric optical depth and turbidity,(WMO TD No. 659), December 1993; Chapter 4: Working Group Discussions – Sunphotometry (pp. 4-5)"

- "WMO/GAW report, Global Atmosphere Watch measurements guide, WMO/TD- No. 1073; GAW Report- No. 143, 2001" -> Specify chapter and use the unique WMO/GAW citation style: "WMO/GAW report No.143, Global Atmosphere Watch measurements guide, WMO/TD- No. 1073; 2001; Chapter 3: Aerosol and Optical Depth (pp. 33-49)"

⇒ Please Sort the WMO/GAW reports by GAW report number in the references list.

- (4.3) Put in the reference list the citation of Michalsky et al. 2001 -> I guess: "Michalsky JJ, Schlemmer JA, Berkheiser WE, Berndt JL, Harrison LC, Laulainen NS, Larson NR, Barnard JC. Multiyear measurements of aerosol optical depth in the Atmospheric Radiation Measurement and Quantitative Links programs. Journal of Geophysical Research: Atmospheres. 2001 Jun 16;106(D11):12099-107."

- (13.27) Harrison and Michalsky (1994) -> This citation is not the correct one and is not in the references list. I guess you want to cite [Harrison et al. ,1994] (Harrison, Michalsky and Berndt, Appl. Opt. 1994)

- (14.4) Please put in the references list the reference of "Smirnov et al. 2000" -> I guess: "Smirnov A, Holben BN, Eck TF, Dubovik O, Slutsker I. Cloud-screening and quality control algorithms for the AERONET database. Remote Sensing of Environment. 2000 Sep 30;73(3):337-49."

3.2. Mathematical formulae (equations):

The quality of the formulae has to be improved. If you use a parameter terminology in a formula it has to be defined in the formula block or in the text above or below. Do not hesitate to write more formulae in order to help the reader to follow the mathematical reasoning.

- Equation 1: In the current version of the manuscript, the paragraph introducing equation 1 (2.32 – 2.37) is unclear. I suggest you to cite before Beer Lambert in the atmosphere (transmission = exp(-(tau_aer + tau_rt))) and to write the equation T = I/I0, then only write the equation (1) as a consequence of the 2 others.

- Equation 3: please explain each term used in the equation. Is Nref the number of referent instruments (in this case of a triad Nref = 3)? What is the origin of the factor 1.96?

- In the text (9.32) You mention the average values <V_0xR> (in the text with a bar for average). Is it an average over the days? Over the number of measurements?

3.3. Other technical comments

- (3.4) The origin and computing of U95 is unclear. Can you repeat the GAW/WMO rules in the text and give a citation from a publication or a GAW report explaining U95 in detail?

- Figure 1: The legend of the left picture is not readable (the points that specify the colors of the wavelengths are to small)

- (5.14). When you talk about uncertainties, please precise if you are discussing an absolute or a relative uncertainty. This would help a lot the reader who tries to follow the reasoning

- (6.1-3). It is hard to understand the relation between delta_AOD_V0 and de-lata_ln(V0), maybe one equation more would help

- (9.15.) Please cite the WMO report and chapter of the "WMO criteria for AOD inter-comparison"

- (10.10) "in addition we have calculated the V0_U95...". But what is shown on the Figure 6 under the denomination "U95(%)"? Is it V0_U95? Is it CV? Is it something else?

- (14.3) "AOD > 2" -> I guess it is AOD[500 nm]?

- (15.4) "AOD(lambda1) > AOD(lambda1)" -> "AOD(lambda1) > AOD(lambda2)"

---

## Author Comment (AC1) · 4 Dec 2017

We would like to thank Dr. Doppler for his comments and help on improving the manuscript.

SPECIFIC COMMENTS

*- Figure 3: Comparing the uncertainty bars and the slopes, this figure does not show an improvement of the quality of the calibration, when WORCC moved the calibration site from Davos to Izaña. To which is IZO a site of better quality for Langley calibration than PMOD-WRC?*

Figure 3 is not an attempt to compare the Davos and Izana sites in terms of Langley potential "quality". Figure 11 shows the monthly variability for both sites and it is clear that with the exception of the Izana dust intrusion months, Izana shows much lower AOD variability. Back to Fig.3, bars from the Davos period is not directly linked with Langleys but is a mix of Langley transfers and comparison with existing instruments, while the Izana period is based purely on Langleys at the site.

We have investigated the Langley quality at Mauna Loa and Izana in a forthcoming work. Using 15 years of sun-photometer measurements at both sites we concluded that the effect of the aerosol variability at each of the two sites in the Vo determination uncertainty is 0.3% and 0.5% respectively. These percentages are directly the standard deviation of the Vo distribution of all (15 year's) Langleys for each site.

*- Chapter 3.1. – Instrument calibration: In the case of a calibration against the triad. What is the strategy? Do you try to have a representative panel of airmass and AOD or do you prefer to focus on low AOD in order to test the sensitivity for low values (avoid negative values, improve detection of AOD for low aerosol masses)*

A text was added to the document:

"In practice, when an instrument is calibrated against the triad, the only limitation on using the synchronous signals is the cloud presence. So no air mass or AOD limits are included."

*- About criteria for the statistics:*
*For monthly statistics: "A minimum of 30 hourly values is required" -> implies 2 days of measurements are enough if they are full and in summer. Is it reasonable ? Don't you want to introduce a criterion of amount of days per month?*

Yes there is a criterion for days per month also that was added now in the text:

"A minimum of 30 hourly values and 10 days per month are required to calculate the monthly mean."

However it has to be noted that monthly values are not officially submitted to any database so the limits could depend on the potential application that these monthly means are used.

*No criterion of repartition of the minutes during the day. My question about the daily mean: Maybe for one site we have, because of the clouds or the availability of the horizon (mountains), a morning average and in other places an afternoon average - Would the comparison of the daily AOD at these 2 sites be still pertinent?*

No there is no criterion of repartition of the minutes during the day.
In cases that there are instrument horizon problems in one site then there can not be any repartition as blocked by mountains measurements will be always out of the L3 data.
In the case of a site with consistent cloudy conditions in specific parts of the day (e.g. morning): We think that since AOD can be measured only in cloudless conditions at the particular site mean AOD have to be calculated only by the non morning parts of the day that are cloudless, even if there is a certain daily pattern for this. That is because, for example if someone would like to study the aerosol radiative forcing for the site he/she can not use the morning (cloudy) measurements anyway, so AOD retrieved from the rest of the day is the representative AOD for the site.

Other Specific comments
*- In the introduction (1.21), you cite "AOD has been measured with the use of sun-photometers for more than 50 year (Holben et. al., 1998)". I have two comments to this:*
*o Holben et al. 2001 (also in references' list) describe better and longer the 50 years long history of AOD measurements with sunphotometers than Holben et al. 1998*
*I really suggest you to briefly describe this 50 years story of sunphotometers, and more expansively than Holben et al. did. All the authors of this manuscript are staff members of PMOD-WRC, a very historical institution, this is why, the reader expects from you that you have the ambition and motivation to make this historical description by your own. You can cite Volz (1959, 1969), Flowers, Shaw (1976, 1982), Leiterer and Schulz (wmotd 222, 1988) And maybe more recent articles describing long time series at specific sites (Weller and Gericke [Met. Zeit. 2005] for MOL-RAO Lindenberg, Barreto et. al [AMT 2014] for Izana.*

A paragraph has been added
Atmospheric extinction of sunlight has been studied at least since 250 year ago (P. Bouger). Linke (1942) turbidity, Angstrom (1929) extinction power law and Junge (1952) with the relationship of particle volume and aerosol number size distribution have mainly set the theoretical basis on studying aerosol extinction. However, Volz (1959) have developed a sun photometer able to measure atmospheric turbidity in different wavelengths using filters, used in the first (U.S.A) (Volz, 1969) and the first European Flowers (1969), Network of turbidity measurements. Since then various sites have included AOD measurements to their monitoring schedule constructing long term series of AOD (e.g. Barreto et al., 2014, Weller and Gericke, 2005, Nyeki et al., 2012). Most of these measurements are site-specific, with little relevance to long term trend analysis on a global scale, however, more recently, several multi-year spatial studies (Holben, 2001; Che et al., 2015, Mitchell et al., 2017) have been conducted.

*- In the introduction (2.17), you mention that "GAW-PFR aims to provide inter-comparison information between networks by overlapping sites". -> Is it only an objective (aim) or are there already studies that make inter-comparison of networks? If there are some studies, please mention them and cite the corresponding publications.*

Unfortunately there are not too many studies published. There are actually three studies under preparation for AERONET (Izana and Mauna Loa) and SKYNET but they can still not be cited now.

*- Chapter 3.2. – Other issues (13.13-21): The QM parameter tested is well described. Could you inform about the threshold of spectral shift that your QC politic allows for the spectral shift of the spectral channel?*

In PMOD WRC we actually do not characterize PFR filter information frequently. Measurements on two instruments showed a shift of less than 0.2 nm. Filter specification provide a central wavelength with an accuracy of ±0.7 nm. Such measurements are possible when instrument that are calibrated show Rayleigh scattering related deviations pointing at the direction of filter shifts. Such cases are not observed till now.

*- Figure 10, Page 14: It is well shown how each method detects or not some type of clouds. Could you explain what are your own QC using all these different methods? Which data you keep in the Level 2 or Level 3 of GAW-PFR database and which you flag out because you consider them as cloud observations.*

A paragraph was added:

"It has to be noted that final AOD data produced include all available measurements that have passed the quality control procedures, except the cloud flagging ones. So all reported AODs are available, accompanied by a flag showing if and which one or which combination of cloud flagging criteria have been assigned for the particular one minute measurement. "

*TECHNICAL COMMENTS*

*3.1. Citations / references An effort has to be done concerning the reference citations:*

All references related recommendation have been included in the new manuscript

*3.2. Mathematical formulae (equations):*
*The quality of the formulae has to be improved. If you use a parameter terminology in a formula it has to be defined in the formula block or in the text above or below. Do not hesitate to write more formulae in order to help the reader to follow the mathematical reasoning.*

*- Equation 1: In the current version of the manuscript, the paragraph introducing equation 1 (2.32 – 2.37) is unclear. I suggest you to cite before Beer Lambert in the*

*atmosphere (transmission = exp(-(tau_aer + tau_rt)) and to write the equation T = I/I0, then only write the equation (1) as a consequence of the 2 others.*

Changed

*- Equation 3: please explain each term used in the equation. Is Nref the number of referent instruments (in this case of a triad Nref = 3)? What is the origin of the factor 1.96?*

1.96 is the approximate value of the 97.5 percentile point of the normal distribution used in probability and statistics. 95% of the area under a normal curve lies within 1.96 standard deviations of the mean, and due to the central limit theorem, this number is therefore used in the construction of approximate 95% confidence intervals. Its ubiquity is due to the arbitrary but common convention of using confidence intervals with 95% coverage rather than other coverages (such as 90% or 99%). This convention seems particularly common in medical statistics, but is also common in other areas of application, such as earth sciences and social sciences

There was an error in the formula (a $^2$ was missing)

$$U95 = 1.96 \sqrt{\sum_{i=1}^{N_{ref}} \left(\frac{\bar{V}_{0xR} - V_{0x}}{2\sqrt{N_{ref}}}\right)^2 + \sum_{i=1}^{N_{ref}} \left(\frac{\sigma_{0xR}}{\sqrt{N_{days}}}\right)^2}$$

Where $\bar{V}_{0xR}$ mean calibration constants derived by the reference instrument R and averaged over all comparison days (Ndays). V0x is the final calibration constant calculated from all comparison days ($N_{days}$) and all reference instruments ($N_{ref}$) .

*- In the text (9.32) You mention the average values <V_0xR> (in the text with a bar for average). Is it an average over the days? Over the number of measurements?*

Text added
The average, over the number of measurements over a day, values $\bar{V}_{0xR}$

3.3. Other technical comments
*- (3.4) The origin and computing of U95 is unclear. Can you repeat the GAW/WMO rules in the text and give a citation from a publication or a GAW report explaining U95 in detail?*

Text was added

According to WMO, 2005, as traceability is not currently possible based on physical measurement systems, the initial form of traceabilty will be based on difference criteria. That is, at an inter-comparison or co-location, traceability will be established if the difference between one network's AOD and another's is within specific limits. Those limits for finite field of view instruments have bene set (WMO, 2005) to 0.005 + 0.01/m optical

depths and the acceptable traceability is when 95% of the absolute AODs are within those limits.So requiring 95% uncertainty (U95) within ±0.005 + 0.01/m optical depths, where the first term (0.005) is linked to instrument uncertainties (signal linearity, sun pointing, temperature effects, processing, etc.) and the second term to a calibration uncertainty of 1%.

*- Figure 1: The legend of the left picture is not readable (the points that specify the colors of the wavelengths are to small)*

Corrected

*- (5.14). When you talk about uncertainties, please precise if you are discussing an absolute or a relative uncertainty. This would help a lot the reader who tries to follow the reasoning*

Text added
Based on Eq. 1 the AOD absolute uncertainty, $\delta AOD_{Vo}$ that is related only to the Langley calibration factor equals $\frac{\delta \ln(Vo)}{m}$ where $\delta ln(V_o)$ is the uncertainty in $ln(V_o)$.

*- (6.1-3).*
*It is hard to understand the relation between delta_AOD_V0 and delata_ln(V0), maybe one equation more would help*

An explanation is provided based on the (new) equation 2.

*- (10.10) "in addition we have calculated the V0_U95*

Done

*". But what is shown on the Figure 6 under the denomination "U95(%)"? Is it V0_U95? Is it CV? Is it something else?*

Figure axis text was corrected, it is Vo_U95 in %.

- (14.3) "AOD > 2"  I guess it is AOD[500 nm]?

Corrected

- (15.4) "AOD(lambda1) > AOD(lambda1)" -> "AOD(lambda1) > AOD(lambda2)"

Corrected

---

## Referee Comment (RC2) · Anonymous Referee #2 · 5 Dec 2017

The manuscript: "The World aerosol Optical depth Research and Calibration Center (WORCC), Quality assurance and quality control of GAW-PFR AOD measurements" by Kazadzis et al., presents the AOD calibration for PFR available in the World Optical Depth Research Calibration Center (WORCC) is a section of the World Radiation Center at Physikalisches-Meteorologisches Observatorium (PMOD/WRC), Davos, Switzerland. A proper calibration is important since it defines the accuracy and quality of measurements, and even though the WORCC has been operative for many years there was not an available protocol calibration for PFR, this paper will become a ref-

erence paper when talking about PFR measurements and calibration within the GAW-PFR network. The paper covers the requirements proposed by the journal. There are some aspects of this paper that require attention by the authors before the paper is ready for publication.

Comments:

After the new Authors Comment my main points about the paper are the following:

RIMA network like to use Toledano et al as a reference: Toledano, C., Cachorro, V. E., Berjon, A., de Frutos, A. M., Fuertes, D., Gonzalez, R., Torres, B., Rodrigo, R., Bennouna, Y., Martin, L., and Guirado, C.: RIMA-AERONET network: long-term monitoring of aerosol properties, Opt. Pura Apl., 44, 629–633, 2011 instead of Prats et al 2011.

What is the actual collaboration between the different networks? Most of the networks mentioned in the paper have the Cimel photometer as the main instrument and they follow the AERONET protocols for calibration, the comparison between PFR-GAW network and the others will help with the error estimation and analyze.

The principal point for me is related to the accessibility of the data, looking at the NILU ebas.nilu.no web there are only a few data for most of the stations why that? Is important that you check it and make available the data of all the stations based on what is presented in table 3.

---

## Referee Comment (RC3) · L. Doppler (Referee) · 5 Dec 2017

Thank you Dr. Kazadzis and your co-authors for your very completed answer that answer clearly all my questions and fullfill plently all the requirements that I have set. I am looking forward to read the comments of the second reviewer in order that you can submit your final corrected manuscript that I will be happy to read. Best regards, Lionel Doppler
* * *
[Figure]

https://doi.org/10.5194/gi-2017-51, 2017.

---

## Author Comment (AC2) · 19 Dec 2017

We would like to thank the reviewer for his/her comments.

Comments:
*After the new Authors Comment my main points about the paper are the following:*
*RIMA network like to use Toledano et al as a reference: Toledano, C., Cachorro, V.*
*E., Berjon, A., de Frutos, A. M., Fuertes, D., Gonzalez, R., Torres, B., Rodrigo, R.,*
*Bennouna, Y., Martin, L., and Guirado, C.: RIMA-AERONET network: long-term*
*monitoring of aerosol properties, Opt. Pura Apl., 44, 629–633, 2011 instead of Prats*
*et al 2011.*

Done.

*What is the actual collaboration between the different networks? Most of the networks*
*mentioned in the paper have the Cimel photometer as the main instrument and they*
*follow the AERONET protocols for calibration, the comparison between PFR-GAW*
*network and the others will help with the error estimation and analyze.*

There are various actions including scientists and institutes responsible for the
calibration of instruments towards a framework to achieve homogeneity,
compatibility and harmonization among the different spectral AOD
networks/instruments. Such actions include MoUs among institutes, long term
measurements of reference instruments at common sites (e.g. Mauna Loa and Izana,
Spain, as described in this paper), intercomparison campaigns (e.g. Kazadzis et al.,
2017, in review, https://www.atmos-chem-phys-discuss.net/acp-2017-1105/acp-2017-
1105.pdf).
Concerning WMO–GAW and the World Data center for Aerosols, measurement
traceability and data quality are essential requirements for monitoring atmospheric
aerosol optical properties by International radiometer networks towards their
inclusion in the World Aerosol Data Center (WMO-WDCA). GAW-PFR and recently
SKYNET networks has been included as WMO-GAW contributing networks to
WDCA. The WMO-CIMO (Commission for Instruments and Methods of
Observation) defined the standard AOD reference for such traceability and
comparison actions, that is the PFR triad, as described in the paper. So for example
SKYNET network has started traceability comparison actions (e.g.
http://www.euroskyrad.net/quatram.html) towards such goals.
In general all the above mentioned actions will help on harmonizing datasets, that
could be used in aerosol trends or case studies and on satellite validation aerosol
related research. Also, comparison of different calibration procedures and AOD
processing basics will help to error estimations and accuracy assessment as the
reviewer has mentioned.

*The principal point for me is related to the accessibility of the data, looking*
*at the NILU ebas.nilu.no web there are only a few data for most of the stations why*
*that? Is important that you check it and make available the data of all the stations*
*based on what is presented in table 3.*

WDCA submissions include some basic rules that have to do with the data quality linked with the instrument re-calibration that it is performed up to 2 years after the actual measurement. (More or less like the AERONET final level 2 data) final data are submitted to WDCA after the recalibration of the instrument with a tentative date to be the end of the next year, for each year's data.

In addition, real time data are submitted. This option for GAW-PFR is at the moment not available, but real time data will start appearing again in the WDCA data base, till the end of January 2018.

Excluding two stations that were added to GAW – PFR network this year and data are processed, about 80% of the data mentioned in the table already exist in WDCA. The missing 20% is for the above mentioned reasons or individual station related/ reprocessing reasons or data that are already submitted but they still have not appeared in WDCA site (WDCA data checking phase).

| Station (abbreviation) | Country | PFR AOD Time-Series | ebas | Submitted (Years) | Reason of missing data |
|---|---|---|---|---|---|
| Alice Springs (ASP) | Australia | 2002 – present | 2002-2015 | 14/15 | Processing 2016 |
| Bratts Lake (BRA) | Canada | 2001 – 2012 | 2001-2012 | 11/11 | Ok |
| Danum Valley (MAL) | Malaysia | 2007-2016 | - | 0/9 | Reprocess of all data |
| Hohenpeissenberg (HPB) | Germany | 1999 – present | 1999-2015 | 15/17 | Processing 2015-16 |
| Izana (IZO) | Spain | 2001 – present | 2001-2014 | 13/15 | reprocessing 2015-16 |
| Jungfraujoch (JFJ) | Switzerland | 1999 – present | 1999-2016 | 17/17 | ok |
| Mauna Loa (MLO) | USA | 2000 – present | 2000-2015 | 15/16 | Processing 2016 |
| Mace Head (MHD) | Ireland | 2000 – 2015 | 2001-2007 | 7/15 | Corrections to old files |

| | | | | | |
|---|---|---|---|---|---|
| Ny Ålesund (NYA) | Svalbard | 2002 – present | 2002-2016 | 14/14 | OK |
| Ryori (RYO) | Japan | 2002 – present | 2002-2016 | 15/15 | OK |
| Cape Point (CPT) | S. Africa | 2007- present | 2014-2015 | 2/10 | Corrections to old files |
| Mt. Waliguan (WLG) | China | 2007- present | 2008-2015 | 8/9 | reprocessing 2015-16 |
| Valentia (VAL) | Ireland | 2007 - present | 2008-2016 | 9/9 | ok |
| Marambio (MAR) | Argentina | 2005 - present | | 1/12 | New GAW stations processing data |
| Troll (TRO) | Antartica | 2012-present | | 2/6 | New GAW stations processing data |
| total | | | | 143/190 (75%) 140/172 (81%) | All Excluding. new |